Medical Imaging with Deep Learning – Under Review 2020          Short Paper – MIDL 2020 submission

# Spine intervertebral disc labeling using a fully convolutional redundant counting model

**Author(s) names withheld**                                    EMAIL(S) WITHHELD

## Abstract

Labeling intervertebral discs is relevant as it notably enables clinicians to understand the relationship between a patient's symptoms (pain, paralysis) and the exact level of spinal cord injury. However manually labeling those discs is a tedious and user-biased task which would benefit from automated methods. While some automated methods already exist for MRI and CT-scan, they are either not publicly available, or fail to generalize across various imaging contrasts. In this paper we combine a Fully Convolutional Network (FCN) with inception modules to localize and label intervertebral discs. We demonstrate a proof-of-concept application in a publicly-available multi-center and multi-contrast MRI database (n=235 subjects). The code is publicly available at [URL will be added after the double blind review].

**Keywords:** Deep learning, Keypoints detection, Spinal cord, MRI, Intervertebral disc

## 1. Introduction

Detection and labeling of intervertebral discs is useful in a clinical and academic setting to observe the progression of diseases, or to inform analyses in functional MRI results. Numerous automated detection methods were created to achieve this task. Some are based on template matching, which detects the C2/C3 disc with a HOG-SVM model (Gros et al., 2017) and then finds the following discs with a sliding window that compares with a probabilistic human spine template (Ullmann et al., 2014). Another method is based on a 3D Fully Convolutional Network (FCN) (Chen et al., 2019) that segments the disc and retrieves its center coordinates. However, these methods are sensitive to the variability of MR quality, contrast and resolution. The goals of this study were to (i) adapt an FCN which was shown to work on multimodal CT images for disc segmentation and localization (Chen et al., 2019), (ii) combine the FCN with inception modules to localize intervertebral discs from MRI data and (iii) train the architecture using a publicly-available multi-center and multi-contrast dataset, to strenghten the generalization capabilities of the model.

## 2. Material & method

### 2.1. Data

We used the Spinal Cord MRI Public Database (Cohen-Adad, 2019). This MRI dataset is composed of T2w and T1w data from 235 subjects, acquired at 40 different centers, thereby exhibiting "real-world" variability in terms of image quality. An average of the 6

middle slices of each subject was used as input images to the network. Ground truths were manually-created by defining a single pixel at the posterior tip of each intervertebral disc. The dataset was split into 75%, 10% and 15% for training, testing, and validation.

## 2.2. Preprocessing

3D volumes were preprocessed using the Spinal Cord Toolbox (SCT) v4.0.1 (De Leener et al., 2016). They were resampled at 1 mm isotropic resolution and straightened according to the spinal cord centerline (De Leener et al., 2017) obtained with the spinal cord segmentation (Gros et al., 2019). As part of straightening transformation, the image was cropped to 141x141 pixels around the spinal region. A Contrast Limited Adaptive Histogram Equalization algorithm was applied to reduce contrast variability in the image (Zuiderveld, 1994). To deal with class imbalance, we increased the target size by applying a 10-pixel Gaussian kernel to single-pixel labels.

## 2.3. Processing

Our custom deep learning model based on inception modules (Szegedy et al., 2015) is shown in Figure 1. It extracts several patches within each image, every pixel is therefore processed by the network several times, allowing the model to average over the error and minimize false negatives and false positives, as it was done for counting cells in microscopic slices (Cohen et al., 2017). We trained the network for 1,000 epochs with a combination of dice loss (Milletari et al., 2016), adaptive wing loss (Wang et al., 2019) and L2-loss (squared loss).

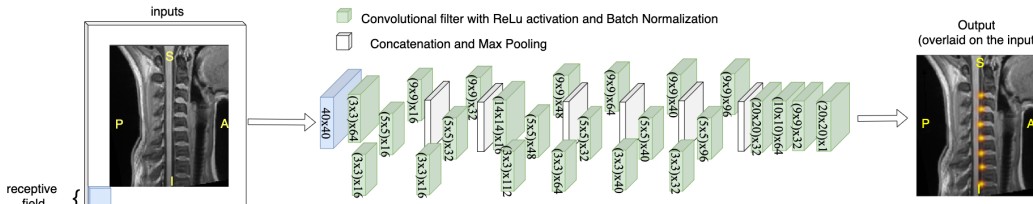

Figure 1: This shows the model with its receptive fields (40x40) on the input. The input is the sagittal view of a T1w MRI and is 0-padded with the size of the receptive field to avoid edge effect. The numbers on the layer represent (kernel size x kernel size) x Number of channels. The output represents the predicted Gaussian functions overlaid on the input from which we will extract discs coordinates.

## 2.4. Metrics

Predicted Gaussian functions were thresholded at 0.5 and the center of mass was retrieved as the predicted coordinates. The performance was evaluated based on the distance between the manually-labeled and the predicted coordinates along the superior-inferior axis as well as False positive rate (FPR) and False Negative Rate (FNR). False positives were defined as predicted points that were at least 5 mm away from any ground truth points or groups of predicted points associated with the same ground truth coordinate. False negatives were counted with ground truth points 5 mm or more away from the predicted points.

## 3. Results

Figure 2 compares our results on the validation set with the previous SCT method using template matching (Ullmann et al., 2014), the ablation study (use of a similar neural network built with inception modules without redundant counting) and the architecture with L1-loss. The proposed model works equally well on the two (T1w and T2w) contrasts, improves prediction precision and reduces the number of FNR and FPR on both modalities. All metrics have been computed based on these methods performance on the validation set.

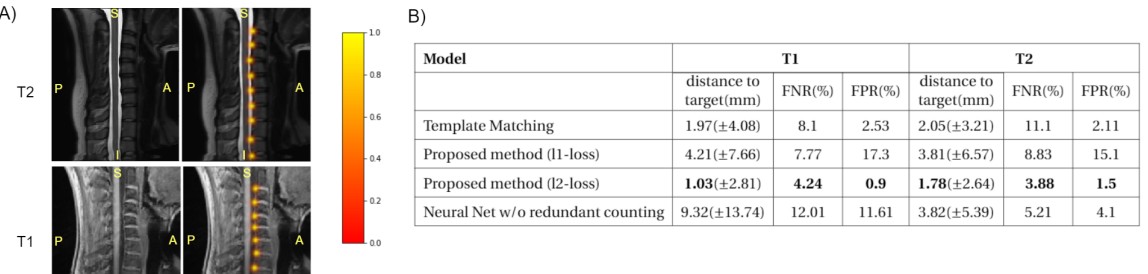

| Model | T1 | | | T2 | | |
|---|---|---|---|---|---|---|
| | distance to target(mm) | FNR(%) | FPR(%) | distance to target(mm) | FNR(%) | FPR(%) |
| Template Matching | 1.97(±4.08) | 8.1 | 2.53 | 2.05(±3.21) | 11.1 | 2.11 |
| Proposed method (l1-loss) | 4.21(±7.66) | 7.77 | 17.3 | 3.81(±6.57) | 8.83 | 15.1 |
| Proposed method (l2-loss) | **1.03(±2.81)** | **4.24** | **0.9** | **1.78(±2.64)** | **3.88** | **1.5** |
| Neural Net w/o redundant counting | 9.32(±13.74) | 12.01 | 11.61 | 3.82(±5.39) | 5.21 | 4.1 |

Figure 2: A) Input images (left) and output predictions of our proposed method (right), color-coded between 0 (transparent/red) and 1 (yellow).
B) Comparison between SCT's template-matching method, the proposed method with L1 and L2 loss, as well as inception network without redundant counting on the evaluation metrics. Smaller distance means better precision. The FNR represents the number of false negatives divided by the total number of ground truth points. The FPR is the number of false positives divided by the total number of predicted points. For both ratio, the smaller the better.

## 4. Conclusion and discussion

This study presents a new architecture for detecting intervertebral discs. The method shows improvement in precision of localization and decrease of false positives/negatives. Future work will include extending the testing of this model to more "real-life" datasets in patients with spinal pathologies (spinal cord injury, multiple sclerosis, tumors, etc.).

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
