# OpenReview forum: "Spine intervertebral disc labeling using a fully convolutional redundant counting model"
_MIDL.io/2020/Conference — Submitted to MIDL 2020_

### Official Review · AnonReviewer4 · 2020-03-10
**Not a new idea but reasonable results.**

**Rating:** 3
**Confidence:** 4

**Review:**

The paper introduces an inception based network to predict gaussians on the posterior intervertebral disks. This problem has important clinical application and the results are reasonable.
There are papers on labeling vertebrae in CT scans and radiographs with very similar approaches. They estimate the vertebrae location and label them using gaussian functions or heat maps. The proposed method in this abstract could be compared to those relevant publications.

1) Payer, Christian, et al. "Integrating spatial configuration into heatmap regression based CNNs for landmark localization." Medical Image Analysis 54 (2019): 207-219.

3) Bayat, Amirhossein, et al. "Vertebral Labelling in Radiographs: Learning a Coordinate Corrector to Enforce Spinal Shape." Computational Methods and Clinical Applications for Spine Imaging: 39.

4) Sekuboyina, Anjany, et al. "Btrfly net: Vertebrae labelling with energy-based adversarial learning of local spine prior." International Conference on Medical Image Computing and Computer-Assisted Intervention. Springer, Cham, 2018.

---

### Official Review · AnonReviewer1 · 2020-03-12
**Labelling IVDs Using FCN**

**Rating:** 3
**Confidence:** 5

**Review:**

The paper presents a method to localise and label intervertebral discs in spinal MRIs. The method takes in a sagittal slice of a T1-weighted MRI and produces gaussian heatmaps of possible disc locations. The paper is a bit vague in terms of how the heatmaps are labelled; do you assume the topmost heatmap is always the C2-C3 IVD or do you predict separate heatmap channels for each IVD?

---

### Official Review · AnonReviewer2 · 2020-03-13
**the method depends on a number of simplifications**

**Rating:** 1
**Confidence:** 5

**Review:**

In the proposed approach, the authors used a fully convolutional network to localize intervertebral discs in MR images. The proposed approach has a number of serious drawbacks.
(1) The proposed approached uses 2D images as the input, that is generated by the average of the six middle slices. This limits the scope of the method, due to the assumption that all centres of intervertebral discs are located in close proximity around the middle slice. It is also not clear what the slice thickness is and thus how the image generated by averaging middle slices looks like.
(2) In the preprocessing step, the 2D image is straightened according to the spinal cord centreline. How is the centreline extracted? If the centreline has been already extracted finding the centres of intervertebral discs and vertebral bodies are greatly simplified and can be done with a variety of approaches that do not require the use of ML-based method.
(3) The term “labelling” of the intervertebral disc might not be correctly used in this work. The proposed network does not distinguish intervertebral disks, but only generates the centres of visible intervertebral disks in the images. The term “localization” should be better.
(4) There is at least one workshop each year at MICCAI that is combined with a challenge dedicated to labelling and segmentation of spine structures. Vertebral body or intervertebral disk localization and labelling is usually part of each completion. Authors should compare their methods with the SOTA method presented in these challenges. Moreover, the CSI workshop at MICCAI 2015 was dedicated to localization and segmentation of intervertebral disks from 3D T2 MRI data.

---

### Official Review · AnonReviewer3 · 2020-03-14
**combination of inception modules and redundant counting for intervertebral disc detection**

**Rating:** 2
**Confidence:** 4

**Review:**

This paper uses a multi-modal, publicly available dataset of 235 patients to detect intervertebral disc coordinates. The paper is well-written and addresses preprocessing and validation metrics. However, additional implementation details could have been included.
1) How was the receptive field size (r) selected? 2) It was not made clear that the network processes sub-images of size r in order to count/detect the discs in the entire image. 3) It is unclear how many patches are extracted within the each image.

Why do the authors think that redundant counting produced better detection?

The detection points on Fig 1. and Fig 2.A. are too small to determine the size of the predicted Gaussian functions.

---

### Meta-Review · Area_Chair1 · 2020-04-08
**MetaReview of Paper277 by AreaChair1**

**Rating:** 2

**Metareview:**

This paper is about intervertebral disc labeling. Reviewers are relatively diverse in their scores. What concerns me is that the authors might have neglected a bunch of related works, particularly from the well-known series of MICCAI CSI workshops. Solid comparisons are necessary in this case to prove the method’s effectiveness.

**Paper Type:**

both

---

### Decision · Program_Chairs · 2020-04-11

Reject